# Contextualising abortion opinions in Kenya: A vignette-based national survey

**Boniface Ayanbekongshie Ushie**[ID][1]*, **Isaiah G. Akuku**[ID][2], **Esther Mutuku**[ID][2], **Kenneth Juma**[2]

**1** Beshi King Development Services, Abuja, Nigeria, **2** Sexual Reproductive Maternal Newborn Child and Adolescent Health, African Population and Health Research Center, Nairobi, Kenya

* boniface.ushie@gmail.com

## Abstract

Abortion is a deeply controversial public health issue, evoking diverse opinions regardless of legal context. Yet policymaking often relies on generalised opinion surveys that lack contextual nuance. We examined public opinions of abortion in Kenya in relation to circumstances of rape, foetal anomaly, and maternal health risk. We used a nationally representative sample of 8,942 adults in Kenya, drawn from a database of 12 million phone users. Using a two-stage sampling approach with random-digit dialling, the sample was stratified by location, sex, and age to ensure proportional representation. A vignette-based questionnaire described three scenarios involving foetal anomaly, threats to the woman's life or health, and rape. It included 14 opinion statements per vignette, each with six response options indicating degrees of agreement or disagreement, and three questions to ascertain levels of support for abortion rights in these circumstances. Trained interviewers administered the questionnaire via telephone between October and November 2022. Multivariable-adjusted linear regression was used to assess factors influencing abortion opinions. Most respondents (> 61%) favoured abortion when pregnancies threaten women's life or health, while only 29% and 44%, respectively, supported it in cases of rape or foetal anomaly. There was significant support for abortion to be performed by the public health system, yet lower support for the idea of abortion as a woman's right. Liberal constructs had higher mean scores than conservative ones, with women's reproductive autonomy scoring highest (3.44 ± 1.07), indicating widespread agreement that abortion decisions should rest solely with the pregnant woman. Among conservative constructs, the sanctity of life had the highest mean score (2.40 ± 0.89), reflecting a strong belief in the value of life among abortion opponents. Abortion opinions varied depending on circumstance, with notable support for legal abortion across contexts. It is recommended that public policy reflect these nuanced views and address key barriers to support, particularly in framing abortion as a matter of public health, gender equality, and human rights.

**Data availability statement:** All data underlying the findings described in this manuscript can be accessed through 10.6084/m9.figshare.30719501.

**Funding:** The research was supported by a grant from the African Regional Office of the Swedish International Development Cooperation Agency, Sida (Contribution No. 12103) to the African Population and Health Research Center (APHRC) under the Challenging the Politics of Social Exclusion project. All authors worked for the APHRC and were supported under the grant. The funders had no role in study design, data collection and analysis, decision to publish, or preparation of the manuscript.

**Competing interests:** The authors have declared that no competing interests exist.

## Introduction

Induced abortion in Kenya is regulated under two main legal instruments: the 2010 Constitution and the 1963 Penal Code. Article 26(4) of the 2010 Constitution specifies that abortion is permitted 'in the opinion of a trained health professional' when emergency treatment is needed, or when the life or health of the mother is at risk, or when it is permitted by any other written law [1]. 'Health', as conceived in the constitutional provision, is not incontrovertibly defined, leaving ambiguities that fuel controversy between abortion supporters and opponents. Abortion opponents prefer a narrower definition of health, while pro-choice groups favour broader definitions that guarantee women's reproductive rights. In 2019, the High Court clarified that survivors of rape are entitled to information about and access to lawful abortion services under these constitutional grounds [2].

Conversely, the Penal Code criminalises abortions under all circumstances, imposing severe penalties on the woman, the provider, and anyone supplying instruments for the procedure [3]. Enforcement and administrative actions under the Penal Code have sometimes intimidated providers and patients seeking abortion-related care. Yet national studies continue to document a high burden of unsafe abortion and abortion-related complications that divert public health resources to post-abortion care [4].

Despite constitutional safeguards, intense debates on the legality of abortion in Kenya continue, with an ongoing legal contestation at the Supreme Court of Kenya centring on the definition of 'health' in the Constitution. The petitioners, opposed to abortion, argue for a restrictive interpretation of health that limits abortion rights primarily to physical health concerns. They also seek clarity on which healthcare professionals are legally authorised to determine eligibility for abortions based on health grounds, since restrictions limit access to care. These debates shape access to services and contribute to uncertainty among both providers and clients [5].

The contention around abortion in Kenya is largely premised on the morality of abortion. Among many, abortion is seen as the killing of unborn human beings and is therefore regarded as morally repugnant. Opponents of abortion argue for the sacredness of life and foetal personification (i.e., they consider the foetus a living being that must be protected) [6]. These moral positions and legal restrictions engender a pervasive social stigma around abortion that manifests as discrimination, insults, harassment, arrests, social exclusion, and denial of services [7]. Under these contexts, many women seeking abortion opt for clandestine yet unsafe methods and procedures.

As part of the legal battle over abortion legality, a public opinion survey commissioned by the Kenya Christian Professional Forum reported overwhelming disapproval of abortion, reflecting a lack of support for abortion provisions in the Constitution [8]. The study reported that 85% of people in Kenya believe 'abortion should never be permitted'. Nevertheless, that study had two notable weaknesses: first, it combined the assessment of people's opinions on three unpopular and controversial topics (i.e., constitutional amendments, homosexuality, and abortion), which might have inherently driven negative responses across the survey. Second, the questions on abortion legalisation frequently drew mixed and contradictory responses

[9], challenging reductionist conclusions. Evidence has shown that people can hold liberal abortion views but oppose the codification of abortion choice in constitutions [9]. A more recent survey conducted in 2024 similarly reported that 88% of Kenyan adults believe abortion should be illegal in all or most cases, reinforcing the dominance of restrictive attitudes in general questioning [10].

Opinions about abortion are complex [11], and community-level perspectives around abortion can be heterogeneous [12]. In many cases, people have contradictory positions and appear to be simultaneously pro-life and pro-choice [9,13]. At the same time, many see no need to clarify these tensions in their positions on abortion, as they believe in both the sanctity of life and the importance of individual choice [14]. Moreover, studies show that negative opinions of abortion often become more nuanced and tempered when the reasons why a woman might seek an abortion are discussed or explained [12]. Cross-national studies in sub-Saharan Africa confirm that public approval is highest for abortions to save a woman's life or health and lower for socio-economic reasons, although acceptance for socio-economic reasons has been increasing in some countries [15–17]. Recent empirical research among very young adolescents (aged 10–14) in Kenya's urban informal settlements illustrates this complexity; while overall attitudes remain conservative, 40–49% of the respondents endorsed abortion in specific scenarios, such as unmarried pregnancy, being too young to parent, or the desire to stay in school, with responses shaped by entrenched gender norms and expectations around heterosexual relationships, although no significant gender differences were observed [18].

Understanding public opinion regarding abortion is critical to the advocacy, legal, and policy processes that surround access to safe abortion and other essential sexual and reproductive health services. The public's position on abortion is often described as 'situationist' – conditioned by the circumstances in which abortions are carried out – rather than 'absolutist' [19]. There is significant public approval of abortions for health and life reasons; however, opinion is divided on abortion for socio-economic reasons [20]. Changes in abortion laws in South Africa, Zambia, and Benin suggest significant transitions, and abortion for economic and social reasons is becoming more acceptable over time [10,11]. However, measuring abortion attitudes remains methodologically challenging. Recent work highlights that using situational vignettes, list experiments, and other indirect methods can reduce bias and elicit more accurate and nuanced responses to questions about abortion [21–23]. Thus, reliance on traditional opinion surveys alone may not capture the complexity of abortion attitudes in Kenya.

Despite advances, Kenya still lacks nationally representative studies that combine robust sampling with context-sensitive measurement of abortion attitudes. This study addressed that gap by employing a nationally representative mobile-phone survey using situational vignettes to assess attitudes towards abortion in three legally and morally salient contexts: foetal anomaly, rape, and threat to maternal life/health. This approach minimises ambiguity, promotes honest reflection, and enables researchers to capture the subtle conditions influencing individuals' perspectives on abortion [24]. Vignettes help to reduce bias when investigating sensitive topics and yield deeper insights into how contextual variables affect public attitudes towards abortion [25], an issue frequently affected by stigma, legal constraints, and social desirability pressures. Employing vignettes represents a robust methodological strategy for addressing these complexities. We position this study within this vignette-based approach, arguing that context-specific measurement offers more policy-relevant insights than non-situationist polling.

## Materials and Methods

### Study design

The study used a cross-sectional design, drawing from a nationally representative sample of participants across Kenya's 47 counties to ensure proportional representation. Eligible respondents were adults aged 18 years or older residing in Kenya, with a functional mobile phone number for telephonic interviews. Based on the reported 15% prevalence of support for abortion [8], and assuming 80% power to detect meaningful differences in prevalence across subgroups, a 95% confidence interval, and a 1% margin of error, we estimated a sample of 8,912, after adjusting for 5% non-response.

This sample was recruited from GeoPoll's database. GeoPoll is a mobile phone-based survey provider with a database of over 12 million contacts at the time of the survey. The database of active phone users was, by default, organised by age, gender, and location (county), facilitating a stratified sampling approach.

Phone numbers were randomly selected for each targeted county and run through software to eliminate duplicates, ensuring that only unique phone numbers were retained in the final sampled database. Respondents were recruited via a random-dialling system, with randomisation based on age, sex, and county of residence. Initially, 50,000 mobile numbers representative of age, gender, and location were randomly extracted. Numbers from this pool were systematically contacted through randomised outbound calling until the pool was exhausted. If the desired sample size by specific strata was not achieved with this initial pool, a new representative pool of 10,000 numbers was extracted, and the process was repeated until the predetermined sample size was reached.

The data were acquired between 12 September and 1 November 2022. By the end of data collection, we had successfully interviewed 8,942 individuals, exceeding the target sample by 30. We implemented a callback procedure for unanswered calls, redialling after four hours to prevent excluding potential participants and ensure everyone had an equal chance to participate. If a selected respondent was unreachable or declined participation, we replaced them with a randomly generated phone number from the same pre-selected pool. Overall, 141,975 phone numbers were dialled for this study, with several thousand eliciting provider-automated messages (60,542; 42.7%), being unanswered (27,160; 19.1%), or disconnected/out of service (17,783; 12.5%). Others (21.3%) did not meet the inclusion criteria; 7.7% (10,916) picked up but did not want to participate in any survey; and 5% (6,140) answered the call but were unwilling to participate in this specific survey. Accounting only for those who were unwilling to participate in this specific survey, we estimated that for every 1.5 respondents interviewed to achieve the sample of 8,942, one potential respondent declined to participate.

## Measures

We aimed to assess Kenyan public opinion of legal abortions in cases of foetal anomaly, threats to women's life or health, and rape. Foetal anomalies were included because, with advancements in medical technology, especially improvements in intrauterine scans, the early detection of foetal anomalies has become possible, and the Kenyan Parliament has begun debating it as a potential legal indicator, albeit without resolution. The issue of foetal anomaly is therefore a relatively new consideration in abortion-related debates and could provide additional insights for enhancing safe abortion care. To overcome the limitations of 'non-situationist' questions – direct questions without accompanying contexts [26] – we chose to contextualise our questions. We adopted the Mosaic of Opinions on Induced Abortion questionnaire [27], which uses vignettes. Following each vignette, respondents were presented with questions and opinion statements (see S1 Text).

Cacique et al. [27] used this instrument with healthcare providers in Brazil to assess their support for abortion rights and willingness to provide abortion care for women in different scenarios. We modified the vignettes and corresponding questions to fit the Kenyan context. The respondents were read vignettes with hypothetical cases ('Wanjiku' – foetal anomaly, 'Zawadi' – rape, and 'Wawira' – health/life endangerment), each portraying compelling circumstances that brought the experiences closer to the respondents.

Three questions and 14 opinion statements accompanied each vignette. The questions concerned: a) whether respondents were in favour of a woman's right to abortion under the circumstances; b) whether they favoured such a procedure being provided by the public health system; and c) whether they would support the performance of such a procedure (e.g., do something to aid the woman). The opinion statements, each with six response options indicating degrees of agreement or disagreement, were designed to capture respondents' moral reasoning about abortion. Responses were then categorised into conservative and liberal constructs based on the predominance of pro-choice or pro-life perspectives revealed in responses (see S1–S7 Figs). Cacique et al. [27] grouped the constructs into eight domains: four conservative and four liberal, as shown in Table 1. The conservative constructs into which responses were fitted are the Psychosocial Repercussions of Abortion (PRA), Conservative Emotional Appeal (CEA), Sacredness of Life (SOL), and Conservative Deontology

**Table 1. Constructs of moral reasoning related to abortion.**

| Constructs | Definition | Number of items |
|---|---|---|
| **Conservative constructs** | | |
| PRA | Anti-abortion stance, appealing to the possible psychological after-effects of abortion or the familiar approval or disapproval of having an abortion. | 3 |
| CEA | Use of shocking expressions and images (like 'murder' and 'cruelty') or the equating of the foetus or embryo with a born child. | 3 |
| SOL | The argument that abortion is always morally reprehensible, either because human life is sacred (even in the very early stages), or because the foetus is a potential person. | 6 |
| CD | The argument that parents have a moral duty to protect the foetus, or that abortion is wrong if the woman lies about having been raped (to have access to legal abortion), or if she exhibits any behaviour that could be considered risky for rape. | 4 |
| **Liberal constructs** | | |
| WRA | The argument that women should have the right to decide whether to abort or not, according to their own values and interests. The concept of autonomy is central to this construct. | 5 |
| LEA | Invocation of shocking expressions and images in favour of the right to induce an abortion, like 'torture' and 'assassination' of the mother. | 2 |
| SRR | The argument that the denial of abortion care violates women's fundamental human rights and promotes a public health problem. | 6 |
| FPP | The argument that the human unborn does not have moral status (at least in some circumstances) or that its life is not sacred. This construct seeks a criterion for determining when a living being may be considered to have moral status. | 3 |

*Source: Cacique et al [27].*

(CD). The liberal constructs are Women's Reproductive Autonomy (WRA), Liberal Emotional Appeal (LEA), Sexual and Reproductive Rights (SRR), and Foetal Personhood Problematisation (FPP). (See [27] for a detailed description of these constructs).

As the nomenclatures of these constructs imply, opinion statements reflected a range of views about abortion, appealing to a spectrum of positions. The 14 statements for each vignette used a five-point Likert scale ranging from 1 = 'strongly agree' to 5 = 'strongly disagree', plus a sixth option of 'I do not know'. Seven of the statements supported the right to abortion, and seven opposed it. To ensure consistent directional interpretation across all constructs, items representing liberal positions were reverse-coded so that higher scores consistently indicated greater support for abortion. The variables were computed into eight constructs. Supplemental S1–S6 Figs present the frequency distributions for the 14 items across the three vignettes.

## Independent variables and covariates

We collected information on sex, age, marital status, religion, and education, and adjusted for these variables in the regression analysis. We also considered respondents' knowledge of indications for legal abortion in Kenya as a covariate. Specifically, we asked respondents, 'Do you know if a woman is allowed to have an abortion in Kenya if …?' followed by a list of 14 conditions [28], only three of which constitute legally permissible grounds for abortion in Kenya. These are if the pregnancy poses a risk to i) the health of the woman, ii) the life of the woman, or iii) if the pregnancy results from rape. Knowledge of abortion law was measured by whether respondents correctly answered 'yes' to these three conditions and 'no' to the other 11. Correct responses were coded as '1' and incorrect/do not know as '0' for indicators that are legal in Kenya, with reverse-coding applied for indicators that are not legal. A composite knowledge score was generated by summing the scores, yielding a possible range from 0 to 14, with higher scores indicating greater knowledge. For analysis and reporting, the total score was categorised into three ordered levels: Poor (0–6), Average (7–11), and High (12–14).

Further, we created latent class membership for profiles of opinions on induced abortion using the latent class analysis approach (see S1 Table). We identified four latent classes based on attitudes toward abortion: Legalisation Proponents, Moderate Supporters, Conditional Supporters, and Abortion Opponents, representing 6.8%, 21.4%, 42.5% and 29.3% of the sample, respectively. Model selection statistics for competing two- to five-class solutions are provided in S1 Table.

Although standard information criteria showed incremental improvements as additional classes were added, these gains were small and did not substantially enhance model quality. We therefore retained the four-class solution because it provided an optimal balance between statistical fit and interpretability; specifically, the four-class model demonstrated good class separation (entropy = 0.82) and yielded clearly distinguishable and meaningful attitudinal profiles. By contrast, the five-class solution resulted in lower entropy (0.77) and produced classes that were less distinct and more difficult to interpret. In brief, the legalisation proponents are respondents who favour liberalisation of abortion, i.e., making it legal under all conditions, while abortion opponents are those who oppose abortion under any circumstances.

## Statistical analysis

We summarised continuous variables using means and standard deviations, and categorical variables using frequencies and proportions. Each outcome construct was analysed separately using a multivariable-adjusted linear regression model, with sociodemographic variables specified as independent variables and knowledge of abortion laws and latent class membership specified as covariates. We assessed multicollinearity using generalised variance inflation factors. Model residuals were checked for normality using quantile–quantile plots, and no model misspecifications were identified. We compared means across the latent classes using bootstrapped estimates of means and corresponding confidence intervals, an approach chosen to account for potential non-normality. Missing data were handled using listwise deletion, as the pattern of missingness was deemed completely random. Significance levels for all comparative analyses were set at $p < 0.05$. We analysed the data using R statistical software (Version 4.3.1).

## Ethical approval

The AMREF Ethics and Scientific Review Committee (AMREF-ESRC) reviewed and approved the study protocol and materials (reference number P1240/2022), while the Kenya National Commission for Science, Technology, and Innovation (NACOSTI) granted the research permit (NACOST-NACOSTI/P/22/19653). The IRB approved verbal consent, and as such, all participants gave verbal informed consent, since interviews were conducted via mobile phone. Interviewers dialled selected phone numbers and asked potential participants to position themselves in a quiet location that afforded privacy. Interviewers then introduced the study to the participants, carefully read the study information sheet to them, ensured they understood it, and answered any questions to the participants' satisfaction. Interviewers asked the potential participant whether they consented to participate. If they agreed, the interviewers checked the 'consented' box on the SurveyCTO script on an Android device. Participants received KES 50 (approximately USD 0.50) via M-Pesa (mobile money) upon participation.

## Results

Our sample was evenly distributed by sex. The mean age was 33.1 ± 10.1 years, with most participants aged 35 years or older. Females were slightly older on average than males (mean 34.9 ± 10.5 vs. 31.2 ± 9.4). Significant differences by sex were observed at the educational level, especially for those with no formal education and primary education. Females constituted a higher proportion of those with no formal education than males (68% vs. 32%). Most respondents were in a marital union or relationship (53.6%). Table 2 summarises participants' sociodemographic information, both overall and stratified by sex.

Tables 3–5 summarise respondents' views on abortion when pregnancy has a foetal anomaly (Table 3), results from rape (Table 4), and poses a threat to the life or health of the woman (Table 5).

Global Public Health
PLOS

**Table 2. Characteristics of the survey participants.**

| Characteristics | Overall | Sex | |
|---|---|---|---|
| | N (%) | Male, n (%) | Female, n (%) |
| **All** | 8,942 (100%) | 4,464 (49.9) | 4,478 (50.1) |
| **Interview language** | | | |
| English | 4,787 (53.5) | 2,737 (57.2) | 2,050 (42.8) |
| Kiswahili | 4,155 (46.5) | 1,727 (41.6) | 2,428 (58.4) |
| **Age group** | | | |
| 18–24 | 2,219 (24.8) | 1,201 (54.1) | 1,018 (45.9) |
| 25–34 | 2,640 (29.5) | 1,843 (69.8) | 797 (30.2) |
| 35+ | 4,083 (45.7) | 1,420 (34.8) | 2,663 (65.2) |
| **Age (years)** | | | |
| Mean (SD) | 33.1 (10.1) | 31.2 (9.4) | 34.9 (10.5) |
| Range | 18.0, 85.0 | 18.0, 85.0 | 18.0, 78.0 |
| **Highest education level** | | | |
| None | 228 (2.6) | 73 (32.0) | 155 (68.0) |
| Primary | 1,481 (16.6) | 519 (35.0) | 962 (65.0) |
| Secondary | 3,348 (37.5) | 1,706 (51.0) | 1,642 (49.0) |
| University | 3,882 (43.4) | 2,165 (55.8) | 1,717 (44.2) |
| **Marital status** | | | |
| Partnered | 4,790 (53.6) | 2,303 (48.1) | 2,487 (51.9) |
| Unpartnered | 4,149 (46.4) | 2,159 (52.0) | 1,990 (48.0) |
| **Religion** | | | |
| Christian | 8,303 (92.9) | 4,117 (49.6) | 4,186 (50.4) |
| Other | 636 (7.1) | 347 (54.6) | 289 (45.4) |

A larger proportion of respondents supported women's right to abortion when pregnancies threaten their life or health (70.2%) compared to cases of rape (28.5%). Although abortion because of foetal anomaly is currently not permitted in Kenya, a notable 57.1% of respondents indicated support for women's right to abort for this reason, compared to those in support of abortion in cases of rape (28.5%). Demographically, low proportions of respondents who were interviewed in Kiswahili (31**%**), were female (43.7%), were 35 years or older (41.3%), or who had no formal education (23.2%) supported the right to abortion in cases of foetal anomaly (Table 3). Generally, more respondents preferred abortions to be performed in Kenya's public health system than those who supported women's right to abortion in cases of foetal anomaly (Table 3). Surprisingly, more respondents would take action to help a woman secure an abortion in a case of foetal anomaly (37.5%) (Table 3) than in a case of rape (35.2%) (Table 4). As expected, the proportion of respondents supporting women's right to an abortion when a pregnancy poses a threat to their health or life (60.1%) (Table 5) was higher than the proportion supporting women's right to abortion in cases of foetal defect or rape.

Table 6 summarises the mean scores of the eight constructs. Overall, the liberal constructs had higher mean scores than the conservative constructs. Specifically, women's reproductive autonomy had the highest mean score (3.44 ± 1.07) among all the constructs, suggesting support for women's right to make abortion decisions independently. Among the conservative constructs, the sacredness of life construct had the highest mean score (2.40 ± 0.89), reflecting a strong opposition to abortion among conservatively minded respondents. Conservative deontology showed clustering, with a standard deviation of 0.89, indicating scores were closely centred around the mean.

**Table 3. Frequency distribution of favourability and support for abortion in cases of foetal anomaly, by socio-demographics.**

| Selected characteristics | Favours right to abort in this situation, N = 8900 | | Favours this abortion performed in Kenya's public health system, N = 8898 | | Would do something to support a woman to obtain abortion in this situation, N = 8900 | |
|---|---|---|---|---|---|---|
| | n (%)[1] | p-value[2] | n (%)[1] | p-value[2] | n (%)[1] | p-value[2] |
| **Dominant survey language** | | | | | | |
| English | 2,666 (56.0) | **<0.001** | 3,090 (64.8) | **<0.001** | 2,252 (47.3) | **<0.001** |
| Kiswahili | 1,281 (31.0) | | 1,993 (48.2) | | 1,142 (27.6) | |
| **Age group (years)** | | | | | | |
| 18–24 | 988 (44.8) | **<0.001** | 1,254 (56.8) | **<0.001** | 903 (40.9) | **<0.001** |
| 25–34 | 1,280 (48.7) | | 1,605 (61.0) | | 1,090 (41.5) | |
| 35+ | 1,679 (41.3) | | 2,224 (54.8) | | 1,401 (34.5) | |
| **Sex** | | | | | | |
| Male | 2,002 (45.0) | 0.2 | 2,617 (58.8) | **0.001** | 1,781 (40.0) | **<0.001** |
| Female | 1,945 (43.7) | | 2,466 (55.5) | | 1,613 (36.2) | |
| **Highest education level** | | | | | | |
| None | 53 (23.2) | **<0.001** | 91 (39.9) | **<0.001** | 54 (23.8) | **<0.001** |
| Primary | 406 (27.5) | | 646 (43.9) | | 382 (25.9) | |
| Secondary | 1,352 (40.6) | | 1,797 (54.0) | | 1,144 (34.3) | |
| University | 2,136 (55.3) | | 2,549 (65.9) | | 1,814 (46.9) | |
| **Marital status** | | | | | | |
| Partnered | 1,999 (41.9) | **<0.001** | 2,643 (55.5) | **<0.001** | 1,673 (35.1) | **<0.001** |
| Unpartnered | 1,948 (47.2) | | 2,440 (59.0) | | 1,721 (41.7) | |
| **Religion** | | | | | | |
| Christian | 3,674 (44.4) | 0.6 | 4,742 (57.4) | 0.11 | 3,153 (38.1) | >0.9 |
| Other | 273 (43.3) | | 341 (54.1) | | 241 (38.0) | |

[1]Frequency (%).

[2]Pearson's Chi-squared tests.

Examined against the backdrop of the latent classes of respondents on the abortion opinion continuum, legalisation proponents consistently showed higher mean scores (e.g., for WRA, SRR, LEA, and FPP) across all the constructs, suggesting unwavering support for abortion rights irrespective of the circumstances and opposing arguments. On the other hand, abortion opponents had the lowest mean scores across all the liberal constructs, implying opposition to abortion, no matter what the circumstances. (See S1 Fig for bootstrapped mean construct scores and confidence intervals, and S2–S7 Figs for the full item-level profiles for the vignettes that underlie these class differences.) Higher values on the boxplot indicate stronger support for the legalisation of abortion.

The profiles of attitudes were dissected across both conservative and liberal frameworks. Legalisation proponents exhibited the highest median scores on liberal constructs, especially WRA and LEA, signifying strong support for women's autonomy and reproductive rights. Interestingly, while lower, their scores on conservative constructs did not fall to the bottom of the scale, suggesting that some proponents recognise certain conservative arguments while still supporting legalisation.

Conditional legalisation supporters, the largest group, generally had medians above the midpoint across liberal constructs. This group supported the legalisation of abortion with certain reservations, informed by conservative values, as evidenced by their moderate scores on constructs like PRA and CD.

Moderate legalisation supporters showed median scores near the neutral midpoint, indicating ambivalence. Their spread across liberal constructs suggests a measured endorsement of reproductive rights, tempered by conservative considerations, particularly on SOL and CEA, where their support was less pronounced.

**Global Public Health** [PLOS logo]

**Table 4. Frequency distribution of favourability and support for abortion in cases of rape, by socio-demographics.**

| Selected characteristics | Favours right to abort in this situation, N = 8906 | | Favours this abortion performed in Kenya's public health system, N = 8904 | | Would do something to support a woman to obtain abortion in this situation, N = 8818 | |
|---|---|---|---|---|---|---|
| | n (%)[1] | p-value[2] | n (%)[1] | p-value[2] | n (%)[1] | p-value[2] |
| **Dominant survey language** | | | | | | |
| English | 1,800 (37.8) | <0.001 | 2,314 (48.5) | <0.001 | 1,978 (41.7) | <0.001 |
| Kiswahili | 796 (19.2) | | 1,493 (36.1) | | 1,168 (28.6) | |
| **Age group (years)** | | | | | | |
| 18–24 | 723 (32.7) | <0.001 | 1,000 (45.3) | <0.001 | 846 (38.7) | <0.001 |
| 25–34 | 813 (30.9) | | 1,167 (44.4) | | 985 (37.7) | |
| 35+ | 1,060 (26.1) | | 1,640 (40.3) | | 1,315 (32.7) | |
| **Sex** | | | | | | |
| Male | 1,339 (30.1) | 0.049 | 1,992 (44.8) | <0.001 | 1,661 (37.6) | <0.001 |
| Female | 1,257 (28.2) | | 1,815 (40.7) | | 1,485 (33.7) | |
| **Highest education level** | | | | | | |
| None | 30 (13.2) | <0.001 | 67 (29.4) | <0.001 | 63 (28.4) | <0.001 |
| Primary | 250 (16.9) | | 509 (34.6) | | 426 (29.3) | |
| Secondary | 888 (26.6) | | 1,342 (40.2) | | 1,092 (33.1) | |
| University | 1,428 (36.9) | | 1,889 (48.8) | | 1,565 (40.7) | |
| **Marital status** | | | | | | |
| Partnered | 1,207 (25.3) | <0.001 | 1,873 (39.3) | <0.001 | 1,521 (32.3) | <0.001 |
| Unpartnered | 1,389 (33.6) | | 1,934 (46.8) | | 1,625 (39.6) | |
| **Religion** | | | | | | |
| Christian | 2,372 (28.7) | <0.001 | 3,527 (42.6) | 0.5 | 2,902 (35.4) | 0.10 |
| Other | 224 (35.3) | | 280 (44.2) | | 244 (38.7) | |

[1]Frequency (%).

[2]Pearson's Chi-squared tests.

Legalisation opponents, with medians leaning towards the lower end of the scale on liberal constructs, firmly resisted the idea of legalisation, aligning with the conservative stance that emphasises the psychosocial impact of abortion and the sacredness of life. As shown in Fig 1, the presence of outliers among proponents in the conservative construct of SOL indicates a segment within this group that strongly opposes abortion despite generally favouring legalisation, underscoring the complex, non-binary nature of public opinion on abortion. (For more details, see S1 Fig, which shows differences in means across the latent classes using bootstrapped means and confidence intervals.)

Between groups, the boxplot shows that legalisation proponents and opponents displayed polarised views, strongly supporting and opposing abortion, respectively, with proponents favouring liberal constructs and opponents favouring conservative constructs. Conditional and moderate supporters presented more nuanced, middle-ground positions, leaning towards and showing ambivalence about legalisation. The class-level differences are supported by mean construct scores (bootstrapped means and 95% CIs) presented in S1 Fig, and by item-level response profiles for each vignette, which are plotted in S2–S7 Figs. Detailed patterns that underlie the class labels (e.g., how specific items loading on Women's Reproductive Autonomy, Sacredness of Life, or Conservative Deontology differentiate the classes) are presented in S2–S7 Figs for the vignette-specific item-by-item distributions.

Multivariable-adjusted linear regression analysis (Table 7) summarises associations for the liberal constructs and shows that age was a determinant of WRA and LEA. Compared with respondents aged 18–24, those aged 25–34 had slightly

**Table 5.** Frequency distribution of favourability and support for abortion in cases of threat to women's health and life, by socio-demographics.

| Selected characteristics | Favours right to abort in this situation, N=8890 | | Favours this abortion performed in Kenya's public health system, N=8906 | | Would do something to support a woman to obtain abortion in this situation, N=8796 | |
|---|---|---|---|---|---|---|
| | n (%)[1] | p-value[2] | n (%)[1] | p-value[2] | n (%)[1] | p-value[2] |
| **Dominant survey language** | | | | | | |
| English | 3,247 (68.2) | **<0.001** | 3,586 (75.3) | **<0.001** | 3,306 (70.0) | **<0.001** |
| Kiswahili | 2,260 (54.7) | | 2,662 (64.3) | | 2,041 (50.1) | |
| **Age group (years)** | | | | | | |
| 18–24 | 1,367 (61.9) | 0.12 | 1,564 (70.8) | 0.14 | 1,376 (63.1) | **<0.001** |
| 25–34 | 1,668 (63.5) | | 1,875 (71.2) | | 1,706 (65.5) | |
| 35+ | 2,472 (61.0) | | 2,809 (69.1) | | 2,265 (56.5) | |
| **Sex** | | | | | | |
| Male | 2,744 (61.7) | 0.7 | 3,139 (70.6) | 0.4 | 2,750 (62.4) | **0.001** |
| Female | 2,763 (62.1) | | 3,109 (69.7) | | 2,597 (59.1) | |
| **Highest education level** | | | | | | |
| None | 106 (46.9) | **<0.001** | 113 (49.8) | **<0.001** | 84 (38.4) | **<0.001** |
| Primary | 734 (49.8) | | 882 (59.7) | | 683 (47.2) | |
| Secondary | 2,010 (60.3) | | 2,291 (68.7) | | 1,943 (58.8) | |
| University | 2,657 (68.9) | | 2,962 (76.6) | | 2,637 (68.9) | |
| **Marital status** | | | | | | |
| Partnered | 2,898 (60.8) | **0.017** | 3,292 (69.0) | **0.009** | 2,710 (57.6) | **<0.001** |
| Unpartnered | 2,609 (63.3) | | 2,956 (71.5) | | 2,637 (64.5) | |
| **Religion** | | | | | | |
| Christian | 5,152 (62.4) | **0.002** | 5,830 (70.5) | **0.022** | 4,982 (61.0) | 0.2 |
| Other | 355 (56.2) | | 418 (66.1) | | 365 (58.3) | |

[1]Frequency (%).
[2]Pearson's Chi-squared tests.

**Table 6.** Mean summary of scores from the mosaic of opinions on induced abortion constructs.

| Constructs | N[1] | Mean (SD) |
|---|---|---|
| **Conservative constructs** | | |
| PRA | 8,890 | 2.24 (1.08) |
| CEA | 8,886 | 2.24 (1.08) |
| SOL | 8,800 | 2.40 (0.89) |
| CD | 8,810 | 2.18 (0.89) |
| **Liberal constructs** | | |
| WRA | 8,861 | 3.44 (1.07) |
| LEA | 8,895 | 3.42 (1.33) |
| SRR | 8,599 | 3.39 (1.02) |
| FPP | 8,594 | 3.37 (1.07) |

[1]N=Number of observations.

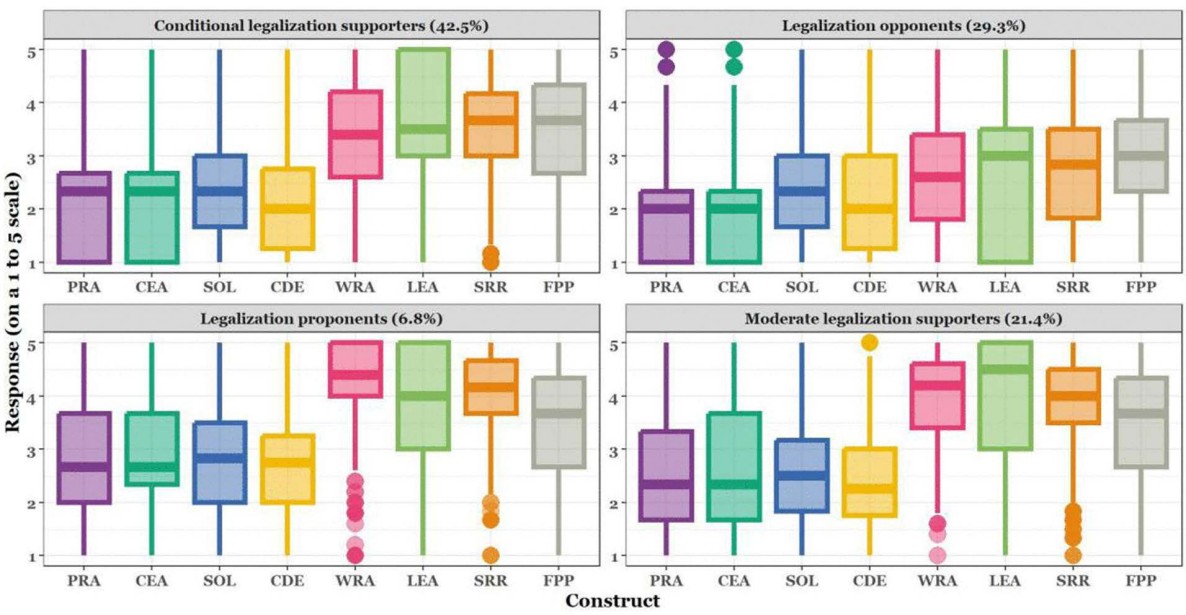

**Fig 1. Distribution of abortion opinions based on liberal and conservative constructs and latent classes.**

lower WRA and LEA scores (WRA: coef = −0.06, SE = 0.03, p < 0.05; LEA: coef = −0.08, SE = 0.04, p < 0.05), showing a modest decrease in these liberal-leaning constructs with age. Sex was a major determinant in all the liberal constructs, with females demonstrating a more favourable stance towards abortion than males. For instance, females scored higher on WRA than males (coef = 0.12; SE = 0.02; p < 0.001), showing a 0.12-point higher WRA score than men. Post-secondary education was associated with moral reasoning for WRA, LEA, and SRR. Knowledge of abortion law was also important; participants with average knowledge scored 0.38 points higher on the LEA construct (coef = 0.38, SE = 0.11, p < 0.001) than those with poor knowledge, while those with high knowledge scored 0.63 points higher (coef = 0.63, SE = 0.18, p < 0.001) than those with poor knowledge.

Sex and age were associated with PRA on the conservative constructs (Table 8). Gender differences were small but statistically significant: females had 0.06 points lower PRA scores than males (coef = −0.06, SE = 0.02, p < 0.05), consistent with males expressing stronger endorsement of psychosocial-repercussion arguments. Similarly, older respondents were significantly associated with the notion of abortion having psychological repercussions for the woman. Furthermore, age was associated with CEA and CD, as the results show that more older respondents opposed abortion using a conservative emotional appeal. Education was associated with lower endorsement of sacredness-of-life arguments; compared with respondents with no education, those with secondary education had SOL scores 0.24 points lower (coef = −0.24, SE = 0.06, p < 0.001), and those with university education had SOL scores 0.18 points lower (coef = −0.18, SE = 0.06, p < 0.01) for this construct.

In CEA, SOL, and CD, knowledge of abortion laws was a major factor; the better the knowledge, the more likely respondents were to support abortion even when arguments using these conservative constructs were deployed.

## Discussion

We examined public opinion of abortion in specific contexts – threat to the woman's health or life, rape, and foetal anomaly – using scenarios based on these indicators. Consistent with previous research showing that abortion attitudes are highly context-dependent and better captured through vignette-based designs, we used a multi-scenario approach to

**Table 7. Multivariable-adjusted linear regression for the determinants of liberal abortion opinions.**

| Variable | Liberal constructs | | | | | | | |
|---|---|---|---|---|---|---|---|---|
| | WRA | | LEA | | SRR | | FPP | |
| | Coef.[1,2] | SE[2] | Coef.[1,2] | SE[2] | Coef.[1,2] | SE[2] | Coef.[1,2] | SE[2] |
| **Age group** | | | | | | | | |
| 18–24 years | Ref. | — | Ref. | — | Ref. | — | Ref. | — |
| 25–34 years | -0.06* | 0.03 | -0.08* | 0.04 | -0.03 | 0.03 | -0.07 | 0.03 |
| 35 + years | -0.05 | 0.03 | -0.02 | 0.04 | -0.01 | 0.03 | -0.05 | 0.03 |
| **Gender** | | | | | | | | |
| Male | Ref. | — | Ref. | — | Ref. | — | Ref. | — |
| Female | 0.12*** | 0.02 | 0.07* | 0.03 | 0.08*** | 0.02 | 0.06* | 0.02 |
| **Highest education level** | | | | | | | | |
| None | Ref. | — | Ref. | — | Ref. | — | Ref. | — |
| Primary | 0.07 | 0.07 | 0.12 | 0.09 | 0.11 | 0.06 | 0.10 | 0.08 |
| Secondary | 0.10 | 0.07 | 0.14 | 0.09 | 0.24*** | 0.06 | 0.13 | 0.08 |
| University | 0.18** | 0.07 | 0.27** | 0.09 | 0.36*** | 0.06 | 0.08 | 0.08 |
| **Marital status** | | | | | | | | |
| Partnered | Ref. | — | Ref. | — | Ref. | — | Ref. | — |
| Unpartnered | 0.07** | 0.02 | -0.01 | 0.03 | 0.02 | 0.02 | -0.02 | 0.03 |
| **Religion** | | | | | | | | |
| Christian | Ref. | — | Ref. | — | Ref. | — | Ref. | — |
| Other | -0.04 | 0.04 | -0.06 | 0.05 | 0.08* | 0.04 | 0.01 | 0.05 |
| **Abortion laws knowledge** | | | | | | | | |
| Poor | Ref. | — | Ref. | — | Ref. | — | Ref. | — |
| Average | -0.02 | 0.08 | 0.38*** | 0.11 | -0.02 | 0.08 | 0.26** | 0.09 |
| High | 0.24 | 0.14 | 0.63*** | 0.18 | 0.14 | 0.13 | 0.36* | 0.16 |
| **Adjusted R²** | 0.235 | | 0.128 | | 0.266 | | 0.042 | |

Note: WRA = Women's Reproductive Autonomy; LEA = Liberal Emotional Appeal; SRR = Sexual and Reproductive Rights; FPP = Foetal Personhood Problematisation.

[1]*p < 0.05; **p < 0.01; ***p < 0.001.

[2]Coef. = Coefficient, CI = Confidence Interval, SE = Standard Error.

Note: Model also adjusted for the dominant survey language (Kiswahili versus English); C1 = Class 1 (Legalisation proponent); C2 = Class 2 (Moderate supporter); C3 = Class 3 (Conditional supporter); C4 = Class 4 (Legalisation opponent); Poor = Poor knowledge; Average = Average knowledge; High = High knowledge; and interaction between knowledge and the latent class.

reveal nuanced situationist views rather than simple pro-choice/pro-life binaries [see [24,29]]. The study revealed that 70% of the participants supported women's right to abortion when pregnancies threatened their life or health; 57% supported abortion in cases of foetal anomaly, and 43% supported abortion in cases of rape. The levels of support for abortion reported in this study were markedly higher than earlier reported for Kenya in 2020 [8], when only 37% indicated support for abortion when the pregnant woman's life is at risk. These findings suggest that contextualising abortion survey opinion questions provides a more nuanced understanding and challenges broad and blanket conclusions that 85% of Kenyans oppose abortion, as the Ipsos [8] survey indicates.

Findings from our study revealed a higher proportion of support for abortion where there is a threat to the woman's life/health, compared to a study in Ghana [30], which recorded only 39% supporting abortion where there is a threat to life/health. Our findings are closer to those of a systematic review of studies among healthcare workers in Ethiopia [31], which reported a pooled favourable attitude towards safe abortion of 66%. On the other hand, our study found lower

**Table 8. Multivariable-adjusted linear regression for the determinants of conservative abortion opinions.**

| Variable | Conservative constructs | | | | | | | |
|---|---|---|---|---|---|---|---|---|
| | PRA | | CEA | | SOL | | CD | |
| | Coef.[1,2] | SE[2] | Coef.[1,2] | SE[2] | Coef.[1,2] | SE[2] | Coef.[1,2] | SE[2] |
| **Age group** | | | | | | | | |
| 18–24 years | Ref. | — | Ref. | — | Ref. | — | Ref. | — |
| 25–34 years | 0.16*** | 0.03 | 0.05 | 0.03 | 0.04 | 0.03 | 0.05 | 0.03 |
| 35+years | 0.22*** | 0.03 | 0.08* | 0.03 | 0.02 | 0.03 | 0.06* | 0.03 |
| **Gender** | | | | | | | | |
| Male | Ref. | — | Ref. | — | Ref. | — | Ref. | — |
| Female | -0.06* | 0.02 | -0.03 | 0.02 | 0.02 | 0.02 | 0.04 | 0.02 |
| **Highest education level** | | | | | | | | |
| None | Ref. | — | Ref. | — | Ref. | — | Ref. | — |
| Primary | 0.06 | 0.08 | -0.04 | 0.08 | -0.17** | 0.06 | 0.02 | 0.06 |
| Secondary | 0.07 | 0.07 | -0.10 | 0.07 | -0.24*** | 0.06 | -0.08 | 0.06 |
| University | 0.13 | 0.08 | 0.06 | 0.08 | -0.18** | 0.06 | -0.03 | 0.06 |
| **Marital status** | | | | | | | | |
| Partnered | Ref. | — | Ref. | — | Ref. | — | Ref. | — |
| Unpartnered | 0.01 | 0.03 | 0.06* | 0.03 | 0.06* | 0.02 | 0.03 | 0.02 |
| **Religion** | | | | | | | | |
| Christian | Ref. | — | Ref. | — | Ref. | — | Ref. | — |
| Other | 0.01 | 0.04 | -0.06 | 0.04 | -0.03 | 0.04 | -0.07* | 0.04 |
| **Abortion laws knowledge** | | | | | | | | |
| Poor | Ref. | — | Ref. | — | Ref. | — | Ref. | — |
| Average | 0.02 | 0.09 | 0.18* | 0.09 | 0.39*** | 0.08 | 0.21** | 0.08 |
| High | 0.26 | 0.16 | 0.81*** | 0.16 | 0.75*** | 0.13 | 0.75*** | 0.13 |
| **Adjusted R²** | 0.067 | | 0.077 | | 0.053 | | 0.053 | |

Note: PRA=Psychosocial Repercussions of Abortion; CEA=Conservative Emotional Appeal; SOL=Sacredness of Life; CD=Conservative Deontology.

[1]*p<0.05; **p<0.01; ***p<0.001.

[2]Coef.=Coefficient, CI=Confidence Interval, SE=Standard Error.

Note: Model also adjusted for the dominant survey language (Kiswahili versus English); C1=Class 1 (Legalisation proponent); C2=Class 2 (Moderate supporter); C3=Class 3 (Conditional supporter); C4=Class 4 (Legalisation opponent); Poor=Poor knowledge; Average=Average knowledge; High=High knowledge; and interaction between knowledge and the latent class.

percentages of support under all conditions than in the United States. Based on a 2018 systematic review, meta-analysis, and analysis of large-scale cross-sectional surveys, Osborne et al. [29] found that 90% of respondents supported abortion if the pregnancy seriously endangers the health of the pregnant woman, 77% approved of abortion if there is a strong chance of foetal anomaly, and 76% approved if the pregnancy resulted from rape. Importantly, Osborne et al. [29] demonstrate that abortion opinions constantly shift and depend on the context or condition. These differences in the prevalence of support for abortion could be linked to the varying study designs, and especially the use of vignettes in our study.

Our findings suggest that there is no clear-cut polarisation of positions into the two extremities of pro- or anti-abortion stances. While overall people tend to gravitate towards one side or the other (those in the middle range of opinions were few, with scores ranging from 2 to 6), people's attitudes are more nuanced and context-dependent than previously assumed. We found that the highest proportion of respondents were those who supported abortion in certain contexts. This finding has implications for advocacy and policy; for example, i) conditional supporters of legalisation could be

targeted for advocacy, and ii) policy reforms could expand access to abortion based on conditionalities if outright legalisation is impossible.

This finding aligns with previous evidence (e.g., 19, 25) suggesting that unidimensional and non-contextual approaches to ascertain abortion-related opinions have limitations because they preclude distinctions among respondents based on salient contextual circumstances. However, like Rominski et al. (2019), we found that the higher the level of education, the higher the likelihood that respondents would support abortion to save a woman's life (68%) [30].

Using conditions already permitted under Kenyan law – threat to a woman's life/health and rape – we found that respondents we classified as abortion opponents were still generally opposed to abortion in such circumstances or had low levels of support. Nevertheless, opposing a woman's right to abortion did not appear to prevent the same respondents from supporting the performance of such abortions in the public health system or taking action to help a woman access abortion in such circumstances. From the existing literature, this finding is not contradictory. For example, Rye and Underhill (2020) identified a group of respondents classified as 'dilemma people' who had negative attitudes towards abortion but had positive attitudes towards choice [9]. Rye and Underhill also recognised a category they termed 'regulated individuals' who are not negative toward abortion but believe that abortion should be strictly controlled rather than an individual choice. Both categories reflect individuals who are supportive of abortion but only to a limited gestational age and/or with additional restrictions, such as obtaining parental or partner consent and institutionalised mandatory waiting periods.

Rye and Underhill's evidence tends to explain otherwise inexplicable findings, in which low proportions of respondents support women's rights to abortion under the three conditions studied, but at the same time, would do something to help a woman to obtain an abortion in such circumstances and would accept the performance of such abortions in the public health system. Further support for this finding comes from a study by Dozier et al. (2020), which found that people can simultaneously believe in the sanctity of life and individual choice [14]. It is imperative to note that respondents falling into these rather conflicted positions typically occupy a position between the polarised pro-choice and pro-life stances. Therefore, we may conclude that, in reality, abortion attitudes are much more nuanced than allowed for in the pro-life/pro-choice dichotomy.

It is also important to note that from our latent class analysis, we identified a group that would support abortion in every circumstance (6.8%) and those who would oppose it in every circumstance (29.3%). Of note are the moderate supporters (21.4%) and the conditional supporters (42.5%). Political leaders rely on public attitudes and opinions to shape policies on critical issues, including abortion, and tend to divide citizens into two opposing camps, whereas the figures show that the majority hold more nuanced views. A proper understanding of the nuances in attitudes could be pertinent, creating guidance for improved access and reduced maternal morbidity and mortality. This study, therefore, reveals relatively low support for rigid pro-choice or pro-life positions among the Kenyan population. Given the current legal ambiguity in Kenya and variable public understanding of statutory requirements, clearer regulations, provider guidance, and public education may translate conditional support into safe access. Evidence from provider studies indicates that accurate legal knowledge among clinicians improves the correct provision of care and reduces informal barriers.

It was notable that 29% of respondents opposed women's right to abortion in in every circumstance, including rape-related pregnancies, even though Kenyan law permits abortion under that condition. This figure reflects a morals-based view of abortion. Religiously minded people argue that one does not know how a child from a rape-related pregnancy may turn out and that children should not be 'killed' or aborted in such cases because it was not their fault that their would-be mothers were raped. This finding hinges on the sanctity of life principle, which holds that taking life and abortion violate God's law and blemish one's spiritual purity [1,20,32]. Therefore, for this group, abortion should not be permitted under any circumstance. People who hold such positions tend to use shocking and expressive imagery that depicts abortion as murder, killing, unholy, and ungodly. However, women living in such social and cultural contexts may be forced into securing clandestine abortions with elevated risks of complications, since restrictive laws do not prevent abortions [4].

Another important finding from this study is the large proportion of respondents who would support women's right to abortion in cases of foetal anomaly (44%), would support such women to secure abortions (38%), and would allow the public health system to perform the procedures (57%). This finding is instructive because it can guide policy reform to expand the conditions under which abortion can include foetal anomaly. Indeed, the process of including such a condition has already been discussed in the Kenyan Parliament, although it has yet to bear fruit.

## Study limitations

This study had some limitations. This was a phone-based survey; only respondents with active phone numbers who were reachable during the data collection period were interviewed. As such, there was a risk of systematically excluding people without active phone numbers and those who could not be reached during the survey. Nonetheless, before the decision was made to exclude any potential participant for being unreachable, several re-dial attempts were made, and they were only excluded if these attempts failed to establish contact. Among telephone owners, we focused on the 12 million phone numbers in GeoPoll's database, implying that millions of people in Kenya with phone numbers but not in the GeoPoll database did not have the same chance of participation. Since abortion is a sensitive topic, social desirability bias and recall bias might have affected the responses. Nevertheless, we assured all respondents of the confidentiality and privacy of their survey responses.

Notwithstanding the limitations, the vignettes provide policymakers and other stakeholders with insights into the diversity of opinions on abortion in the Kenyan context when scenarios are deployed. Other countries with similar legislation may also benefit from similar vignette-based public opinion surveys.

## Supporting information

**S1 Table. Model fit indices for latent class models (2–5 classes).**
(DOCX)

**S1 Fig. Mean construct scores stratified by latent class membership.**
(TIF)

**S2 Fig. Item-level profiles of attitudes toward abortion in cases of foetal anomaly (Statements 1–7: Conservative statements).**
(TIF)

**S3 Fig. Item-level profiles of attitudes toward abortion in cases of foetal anomaly (Statements 8–14: Autonomy- and rights-based statements).**
(TIF)

**S4 Fig. Item-level profiles of attitudes toward abortion in cases of rape (Statements 1–7: Conservative statements).**
(TIF)

**S5 Fig. Item-level profiles of attitudes toward abortion in cases of rape (Statements 8–14: Autonomy- and rights-based statements).**
(TIF)

**S6 Fig. Item-level profiles of attitudes toward abortion when pregnancy poses a threat to the woman's life or health (Statements 1–7: Conservative statements).**
(TIF)

**S7 Fig. Item-level profiles of attitudes toward abortion when pregnancy poses a threat to the woman's life or health (Statements 8–14: Autonomy- and health-rights statements).**
(TIF)

**S1 Text. Questionnaire: Vignettes and accompanying questions and items.**
(DOCX)

## Acknowledgments

We are indebted to the GeoPoll team for setting up the data collection system, supporting field team training, and collecting data. We appreciate the field team's hard work and diligence in collecting the study data. We also thank all the respondents for their participation.

## Author contributions

**Conceptualization:** Boniface Ayanbekongshie Ushie, Kenneth Juma.

**Data curation:** Isaiah G Akuku, Esther Mutuku.

**Formal analysis:** Isaiah G Akuku, Esther Mutuku.

**Funding acquisition:** Boniface Ayanbekongshie Ushie.

**Investigation:** Boniface Ayanbekongshie Ushie, Kenneth Juma.

**Methodology:** Boniface Ayanbekongshie Ushie, Kenneth Juma.

**Project administration:** Boniface Ayanbekongshie Ushie.

**Writing – original draft:** Boniface Ayanbekongshie Ushie.

**Writing – review & editing:** Boniface Ayanbekongshie Ushie, Isaiah G Akuku, Esther Mutuku, Kenneth Juma.

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
