## [Decision Letter · Decision Letter 0]

11 Nov 2025

PGPH-D-25-01893

Contextualizing abortion opinions in Kenya: A vignette-based national survey

Dear Dr. Ushie,

Thank you for submitting your manuscript to PLOS Global Public Health. Two individuals have provided constructive reviews. After careful consideration, we feel that the submitted manuscript has merit but does not fully meet PLOS Global Public Health’s publication criteria as it currently stands. Therefore, we invite you to submit a revised version of the manuscript that addresses the points raised during the review process.

We look forward to receiving your revised manuscript.

Kind regards,

Hannah Tappis, DrPH, MPH

Academic Editor

Journal Requirements:

1. In the ethics statement in the Methods, you have specified that verbal consent was obtained. Please provide additional details regarding how this consent was documented and witnessed, and state whether this was approved by the IRB

1. Please clarify all sources of financial support for your study. List the grants, grant numbers, and organizations that funded your study, including funding received from your institution. Please note that suppliers of material support, including research materials, should be recognized in the Acknowledgements section rather than in the Financial Disclosure.

2. State the initials, alongside each funding source, of each author to receive each grant. For example: "This work was supported by the National Institutes of Health (####### to AM; ###### to CJ) and the National Science Foundation (###### to AM)."

3. State what role the funders took in the study. If the funders had no role in your study, please state: “The funders had no role in study design, data collection and analysis, decision to publish, or preparation of the manuscript.”

4. If any authors received a salary from any of your funders, please state which authors and which funders.

3. In the online submission form, you indicated that All data and materials are available on reasonable request from the corresponding author. Also, according to the APHRC policies (the organization hosting the datasets), all deidentified datasets will be publicly available on the APHRC microdata portal after three years (https://aphrc.org/microdata-portal/).

3. Uploaded as supplementary information.

4. Please provide separate figure files in .tif or .eps format.

5. We have noticed that you have uploaded Supporting Information files, but you have not included a list of legends. Please add a full list of legends for your Supporting Information files after the references list.

Reviewers' comments:

Reviewer's Responses to Questions

**Comments to the Author**

1. Does this manuscript meet PLOS Global Public Health’s publication criteria?

Reviewer #1: Yes

Reviewer #2: Yes

2. Has the statistical analysis been performed appropriately and rigorously?

Reviewer #1: Yes

Reviewer #2: Yes

3. Have the authors made all data underlying the findings in their manuscript fully available (please refer to the Data Availability Statement at the start of the manuscript PDF file)?

Reviewer #1: Yes

Reviewer #2: Yes

4. Is the manuscript presented in an intelligible fashion and written in standard English?

Reviewer #1: Yes

Reviewer #2: Yes

Reviewer #1: Contextualizing abortion opinions in Kenya: A vignette-based national survey

PGPH-D-25-01893

This is a well written paper, and the Introduction suitably makes the case to study public opinions on abortion in Kenya in the context of rape, threat to health and life, and fetal anomalies.

Regarding the study design, it is stated that the database used is maintained by GeoPoll, a mobile phone-based survey provider and that consists of over 12 million contacts at the time of the survey. Do we assume that all service provider customers are included?

The authors should be consistent in the spelling of GeoPoll. Elsewhere it is spelt Geopoll (without a capital ‘P’).

There were 21.3% that did not meet the inclusion criteria, and 5% (6,140) declined to

participate. Are their demographics identified and are they very different from those that participated?

Table 1 aptly summarises the ‘Constructs of the moral reasoning related to abortion’.

The reader subsequently has to constantly refer to that table as to what the abbreviations are. However, in Table 6 (Mean summary of scores from the mosaic of opinions on induced abortion constructs) there is another listing of the constructs and their abbreviations. The reviewer wonders if it may make the article too wordy to state the construct in full, though it is quite conventional to have abbreviations after first defining them.

Note the error in line 166 referring to the reference for Table 1. It is not From Cacique et al (2028), but From Cacique et al (2018). Also, why not place the citation number which is (14).

Noted the results summarised in Tables 3 to 5 (Pearson’s chi squared test) and the reader looks forward to the multivariate analysis which follows. In the narrative related to Tables 7 and 8, it may be useful to use one example from either construct/variable to relate the numbers to the (already stated) narrative in the context of multivariate analysis. Does the Discussion address the significant variables (if any)?

The Supplementary material is very informative, and the authors could do well to lead the reader onto them explicitly in the text.

Noteworthy that Figures S2 to S7 show for the various attitudes (statements 1 to 7 and 8 to 14) were overwhelmingly either ‘You agree too much’ or ‘You agree too little’ with few hedging and even fewer neutral.

For line 347 ......For example, (3) identified a group of respondents classified as “dilemma people”

Suggest ..... For example, Rye and Underhill (year), identified a group of respondents classified as “dilemma people”......... (3).

Similarly, line 357 ......Further support for this finding is a study by (7), which found that people can simultaneously.....

Suggest, Further support for this finding is a study by (authors), which found that people can simultaneously..... (7).

The study limitations are well noted. Nevertheless, even without generalising to the whole Kenyan population, rich data has been captured and presented from the sample that though may have been incomplete (of all phone users) was a representaive sample (of the GeoPoll database).

Notwithstanding the limitations, the vignettes allow policy makers and other stakeholders an insight into a number of types of moral constructs categorised as liberal or conservative related to the scenarios related to abortion in the Kenyan context. Other countries with similar legislation would likely have similar public opinions.

Thank you

Reviewer #2: The methodology adopted in this study is commendably detailed and systematic. Nevertheless, the review of relevant literature appears limited. Considering the growing body of recent research on abortion surveys, the inclusion of additional scholarly works would enhance the robustness of the methodological framework and provide stronger empirical support for the study’s findings.

**Do you want your identity to be public for this peer review?** For information about this choice, including consent withdrawal, please see our Privacy Policy

Reviewer #1: No

Reviewer #2: **Yes:** Oluseun Joshua Adejugbe

---

## [Editor Report · Decision Letter 1]

30 Dec 2025

PGPH-D-25-01893R1

Contextualizing abortion opinions in Kenya: A vignette-based national survey

Dear Dr. Ushie,

Thank you for submitting your manuscript to PLOS Global Public Health. After careful consideration, we feel that it has satisfied our scientific requirements for publication.

However, our editorial team have significant concerns about the grammar, usage, and overall readability of the manuscript. PLOS requires that published manuscripts use language which is 'clear, correct, and unambiguous', see our criteria for publication at https://journals.plos.org/globalpublichealth/s/criteria-for-publication#loc-5. We therefore request that you revise the text to fix the grammatical errors and improve the overall readability of the text.

We suggest you have a fluent English-language speaker thoroughly copyedit your manuscript for language usage, spelling, and grammar. If you do not know anyone who can do this, you may wish to consider employing a professional scientific editing service.

Whilst you may use any professional scientific editing service of your choice, PLOS has partnered with both American Journal Experts (AJE) and Editage to provide discounted services to PLOS authors. Both organizations have experience helping authors meet PLOS guidelines and can provide language editing, translation, manuscript formatting, and figure formatting to ensure your manuscript meets our submission guidelines. To take advantage of our partnership with AJE, visit the AJE website (https://www.aje.com/go/plos/) for a 15% discount off AJE services. To take advantage of our partnership with Editage, visit the Editage website (www.editage.com) and enter referral code PLOSEDIT for a 15% discount off Editage services. If the PLOS editorial team finds any language issues in text that either AJE or Editage has edited, the service provider will re-edit the text for free.

Please note that we will not be able to proceed with publication of your manuscript until the concerns above are addressed.

* A copy of your manuscript showing your changes by either highlighting them or using track changes (uploaded as a supporting information file)

* A clean copy of the edited manuscript (uploaded as the new manuscript file)

We look forward to receiving your revised manuscript.

Kind regards,

Katherine Kokkinias, Ph.D.

Staff Editor

on behalf of

Hannah Tappis, DrPH, MPH

Academic Editor
---

## [Editor Report · Decision Letter 2]

12 Feb 2026

Contextualizing abortion opinions in Kenya: A vignette-based national survey

PGPH-D-25-01893R2

Dear Dr Ushie,

We are pleased to inform you that your manuscript 'Contextualizing abortion opinions in Kenya: A vignette-based national survey' has been provisionally accepted for publication in PLOS Global Public Health.

Best regards,

Hannah Tappis, DrPH, MPH

Academic Editor